# Tissue-Resident Memory T Cells in Cancer Metastasis Control

**DOI:** 10.3390/cells14161297

**Published:** 2025-08-21

**Authors:** Tyler H. Montgomery, Anuj P. Master, Zeng Jin, Qiongyu Shi, Qin Lai, Rohan Desai, Weizhou Zhang, Chandra K. Maharjan, Ryan Kolb

**Affiliations:** 1Department of Pathology, Immunology and Laboratory Medicine, College of Medicine, University of Florida, Gainesville, FL 32610, USA; t.montgomery@ufl.edu (T.H.M.); anujmaster@ufl.edu (A.P.M.); zengjing@ufl.edu (Z.J.); qshi1@ufl.edu (Q.S.); qinlai7@gmail.com (Q.L.); rohan.desai@ufl.edu (R.D.); zhangw@ufl.edu (W.Z.); 2UF Health Cancer Center, University of Florida, Gainesville, FL 32610, USA

**Keywords:** tissue-resident memory T (TRM) cells, cancer, metastasis, cancer therapy

## Abstract

Tissue-resident memory T (TRM) cells have emerged as critical sentinels in the control of cancer metastasis, yet their precise roles across different tumor types and tissues remain underappreciated. Here, we review current insights into the mechanisms governing TRM cell seeding and retention in pre-metastatic niches, their effector functions in eliminating disseminated tumor cells, and their dynamic crosstalk with local stromal and myeloid populations. Here, we highlight evidence for organ-specific variability in TRM cell-mediated immunity, discuss strategies for therapeutically harnessing these cells—ranging from vaccination and checkpoint modulation to chemokine axis manipulation—and explore their promise as prognostic biomarkers. Finally, we outline key knowledge gaps and future directions aimed at translating TRM cell biology into targeted interventions to prevent and treat metastatic disease.

## 1. Introduction

Metastasis remains one of the largest challenges in modern cancer therapy, accounting for the vast majority of cancer-related mortality [1]. Despite this burden, our understanding of how the immune system responds to and controls metastases is still emerging. Tissue-resident memory T (TRM) cells have emerged as a distinct subset of memory T lymphocytes that are critical in tumor control and primary players in immune-checkpoint inhibitor function. Early evidence suggests that TRM cells may also play an essential role in the prevention and control of metastases. TRM biology has been heavily studied in the two decades since its identification, but its involvement specifically in metastatic seeding and progression has yet to be fully synthesized.

This review summarizes current knowledge on how TRM cells influence cancer metastasis, highlighting their core biology, metastatic control program, and clinical significance as both a prognostic biomarker and potential therapeutic target. We also discuss unanswered questions and future directions necessary to better understand TRM cell-mediated metastasis control.

## 2. TRM Biology Relevant to Metastasis Control

TRM cells are a non-circulating subset of memory T lymphocytes (the other known subsets are effector memory and central memory T cells) that take up long-term residence within the tissue of peripheral organs. During a tissue infection, TRM cells elicit a rapid response to eliminate infected cells to evade a potential systemic immune reaction. Traditionally, TRM cells have been identified by their expression of CD69—an early activation marker that inhibits sphingosine-1-phosphate receptor-1 (S1PR1) to prevent tissue egress—and CD103, an integrin that binds epithelial E-cadherin to promote retention within epithelial layers [2]. Despite this, multiple studies have now shown that nearly all TRM markers, including CD69 and CD103, can vary substantially in expression by tissue type, activation state, and even disease-specific context [3,4,5,6]. Here, we limit our discussion to the emerging aspects of TRM biology most relevant to metastasis control. For a comprehensive general overview of TRM cell lineage commitment, full marker panels, and core transcriptional programs, readers are referred to several recent reviews [7,8,9,10].

Below, we focus on three key aspects of TRM biology—tissue seeding, on-site function, and microenvironment influence—that may dictate their roles in metastatic and pre-metastatic tissues and might involve relevant target pathways for future therapies (Figure 1). We then address direct mechanistic evidence demonstrating TRM cell-mediated control of metastasis in preclinical and clinical models.

### 2.1. Tissue Seeding and Retention of TRM Cells

The mechanisms governing how TRM cells differentiate and seed within tumors and pre-metastatic tissues remain incompletely defined, but chemokine and cytokine cues clearly shape homing, local differentiation, and durable residence. CXCR6 is a chemokine receptor and is often expressed by TRM cells and other T cell subtypes, and its roles in immunosurveillance and tissue residence are multifaceted [11,12]. Its ligand, CXCL16, can be expressed by cancer cells and antigen-presenting cells (APCs), and the interaction of the two has been found to play a key role in recruiting and retaining TRMs in various tissue contexts (Figure 2) [12,13]. For instance, tumor-specific CXCR6 knockout (KO) T cells demonstrate greatly reduced TRM cell retention within ovarian tumors, increasing responses in blood and spleen, but reducing resident memory behavior at the primary tumor and leading to poor tumor control [14]. This suggests that CXCR6 is critical for tissue residence.

More studies have focused on the exploration of modulating CXCL16 rather than targeting CXCR6 for exerting influence on TRM cell seeding. Viral-infection models have demonstrated that epithelial-cell-derived CXCL16 establishes a chemotactic gradient to pull CXCR6+ TRM cells into the airway lumen, which can be depleted by interfering with the CXCL16-CXCR6 axis [11]. In cancer, CXCL16 may be directly targetable as monotherapy and as a therapeutic adjuvant. For example, anti-CXCL16 blockade in mouse breast cancer liberates CXCR6+ effectors from primary tumors, driving their migration and differentiation into lung TRM cells (see Section 3) and highlighting a promising target for TRM-directed therapies [15]. In Head-and-neck and lung tumor models, cancer vaccine immunization can drive local CXCL16 upregulation, CXCR6 expression on TRM precursors, and robust TRM recruitment, while impaired TRM cell seeding was observed in CXCR6-deficient mice [16]. Together, these studies indicate that local CXCL16 expression in tissues by stromal or tumor cells plays a critical role in TRM attraction and residency via CXCR6 and may compete to dictate where TRM precursors localize. Moreover, emphasis on balanced interference with this axis is required when modulating homing into pre-metastatic niches and protecting against metastatic seeding.

Complementing this positional axis, transforming growth factor beta (TGF-β) signaling has been identified to promote T cells to adopt tissue resident phenotypes like CD103 expression and subsequent tissue retention (Figure 2) [17]. Loss of TGF-β receptor signaling sharply reduces accumulation of CD103+ TRM-like cells in pancreatic tumor models [18]. Like with CXCL16 blockade, TGF-β neutralization has also been demonstrated to cause downregulation of tissue residency markers, leading to enhanced systemic tumor immunity, secondary tumor response, and overall survival when combined with PD-L1 blockade in mice [19]. This evidence suggests that, like the CXCR6-CXCL16 axis, TGF-β modulation could offer an opportunity to influence TRM behavior to promote systemic tumor immunity and control metastasis, especially when combined with other treatments.
Figure 2Retention of TRM cells at a primary tumor site is influenced by TGF-β and CXCL16 signaling. (**Left**) CXCL16-CXCR6 axis [12,13] and TGF-β signaling [17,19] promote primary tumor retention of TRM cells, while (**Right**) disruption of these pathways has been shown to induce dissemination of TRM cells into distant tissues for metastatic control [15,18,19].
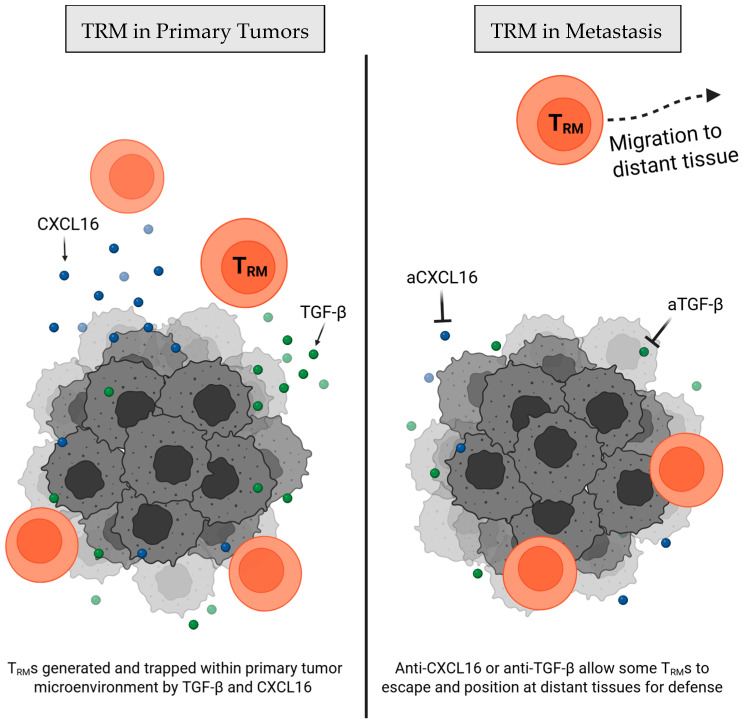


After establishing within target tissues, TRM cell survival depends on interfacing with a dedicated niche. In mouse melanoma models, CD11c+ dendritic cells (DCs) were found to closely cluster with CXCR6+ TRMs around hair follicles, express high CXCL16, and were required to maintain TRM cell populations [20]. Short-term DC ablation disrupted these clusters, and prolonged depletion caused them to disappear completely [20]. Other murine melanoma models showed tumor DCs presenting CXCL16 and trans-presenting IL-15 to incoming T cells in the TME, while blocking these signals led to declines in their survival [21]. While not limited to solely TRM cells, this study suggests a potential role of DC in maintenance of TRM cell populations by directly supplying CXCL16 and IL-15 signals—known to be key to TRM cell survival [21,22]. Further investigation will be needed to understand how this potential interaction between DCs and TRM cells occurs in tissues not harboring active tumors, including pre-metastatic niches. Consideration of the factors that hold and maintain TRM cell populations at tumor-relevant sites will be critical for the design of functional and lasting TRM-based therapeutics.

### 2.2. On-Site TRM Cell Effector Function

TRM cells not only seed and persist in tissues but also exhibit unique functional and metabolic programs that distinguish them from circulating T cells and enable them to endure and act in hostile microenvironments, such as tumors or metastatic sites. TRM cells are distinguished by their ability to launch rapid, high-potency effector responses upon antigen encounter, an attribute that makes them especially efficient killers of tumors [23,24]. Consistently, tumor co-culture experiments have shown that TRM cells demonstrate superior tumor killing when compared to peripheral blood CD8+ T cells [25].

In order to carry out their heightened capability for targeted killing, TRM cells must survive long-term in nutrient-restricted and potentially hypoxic tissue niches, conditions that circulating T cells, which favor glycolysis, are poorly equipped to endure. To accomplish this, some TRM cells have been observed to rewire their metabolism in a tissue-dependent manner. In the lung, TRM cells take up exogenous fatty acids and rely on carnitine-palmitoyl transferase 1A (CPT1A)-mediated B-oxidation for energy [26]. Genetic or pharmacologic inhibition of CPT1A markedly reduces CD103+CD69+ TRM pools in the lung [26]. Similarly, gut TRM cells use fatty acid oxidation to fuel their homeostatic proliferation and survival, the blockage of which leads to TRM cell attrition in the intestine [27]. Beyond lipid oxidation, TRM cells in the small intestine have been found to depend on de novo cholesterol synthesis via the SREBP2-regulated mevalonate pathway. Murine models have demonstrated that this pathway—and downstream coenzyme Q production—is essential for maintaining mitochondrial membrane potential and reactive-oxygen-species buffering in TRM cells [28]. Disruption of these features impaired TRM cell persistence and function without affecting circulating memory cells [28].

It is evident that tissue-dependent metabolism is a core part of TRM cell function, and that consideration of local microenvironment cues—including oxygen and metabolite availability—will be required for the development of TRM cell-based therapies, especially those hoping to prophylactically position TRMs to protect tissues against incoming metastasis. For applications aimed at controlling existing metastatic growths, direct metabolic profiling of TRM cells in metastatic tumors remains unexplored, although some primary tumors have been profiled [29]. Therefore, more work will need to be done to understand the specific TRM cell metabolic requirements in diverse metastatic and pre-metastatic environments.

### 2.3. TRM Cell and Microenvironment Crosstalk

Beyond direct killing, TRM cells serve as on-site immunological alarms, shaping the local tissue by creating an inflamed environment, promoting local T cell responses, and crosstalk with other immune cells to reinforce local and systemic antitumor responses. Mechanistic murine studies have shown that TRM cell activation triggers a cascade of crosstalk, licensing migratory dendritic cells via IFN-γ/TNFα to prime systemic circulating T cells [30] and secreting IL-2/IL-15 to recruit bystander T cells into tissues [31]. Extending these principles to human metastases, single-cell transcriptomics, and multiplex imaging of ovarian omental lesions revealed TRM cell-like clusters co-localized with granulysin+CD4+ T cells, proliferative plasmablasts, and activated macrophages, creating an intensely inflamed immune niche [32]. In viral models, TRM activation in the mouse brain was found to lead to TRM-directed local immune activation and recruitment of immune cells, further supporting their immune-directing capabilities [33]. The authors note that this feature could be harnessed to target brain tumors [33], suggesting a clear therapeutic avenue to be explored to target brain primary tumors and metastases in a TRM-mediated manner. These examples of multicellular immune collaboration illustrate how TRMs can orchestrate a local and systemic immune response that could underpin durable containment of metastases. TRM can therefore be understood as not only direct defenders but also on-site alarms which ignite local inflammation, recruit other immune cells, and promote systemic CTL responses upon antigen re-encounter. As such, TRMs are uniquely equipped and positioned to detect and contain potential metastases, thereby preventing their growth at sites distant from the primary tumor.

## 3. TRM Mechanistic and Phenotypic Insights

Recent studies have overturned the notion of tissue-resident memory T (TRM) cells as mere local sentinels, revealing them instead as active defenders against metastatic initiation and spread. Far from passive residents, CD8+ TRMs infiltrate pre-metastatic niches, intercept circulating tumor cells, and orchestrate a local alarm system that identifies and contains micrometastases. Mechanistic work and observations in both animal models and patient samples suggest that TRM cells may actively patrol and defend metastatic niches; understanding how they carry out these functions will be critical for designing TRM cell-targeted therapies against metastasis.

Tumor-specific TRM cells preemptively position themselves within non-lymphoid tissues to prevent metastatic seeding. In a murine 4T1 breast cancer model, pre-metastatic lungs were found to contain TRM cells with high clonotypic overlap between primary tumor T cells 3 weeks following tumor formation [15]. Further, promotion of T cell egress from the primary tumors using anti-CXCL16 by the same authors enhanced protection against lung metastasis, likely due to the boosted lung TRM cell populations, and significantly reduced metastatic lung tumor burden in a TRM cell-mediated manner [15], suggesting the ability to functionally direct TRM cell populations. By contrast, in metastatic melanoma, lung metastatic seeding was found to be predominantly restrained by circulating memory CD8+ T cells—with little contribution from tissue-resident subsets—highlighting that TRM-mediated metastasis defense may vary markedly by cancer type even within the same organ [34]. Strikingly, targeted prime-boost vaccination with an intramuscular DNA prime followed by an intranasal influenza-vector was demonstrated to overcome this difference, converting circulatory central memory T (TCM) cells to lung-resident CD8+ TRM cells, and that alone sufficed to block melanoma and mesothelioma lung metastases in mice [35]. These data demonstrate that actively directing TRM cell seeding and accumulation offers a powerful strategy to fortify pre-metastatic niches against tumor spread.

In many tissue contexts, TRM cells show significant heterogeneity and multiple distinct populations of different origins. Despite this, the effect of tissue environment on anti-metastasis activities of TRM is just beginning to be investigated. In colorectal cancer liver metastases isolated from human samples, two main subsets of TRM cells were identified as CD103+CD69+ or CD103-CD69+ TRM [24]. Since liver TRM normally develop without TGF-β (and therefore lack CD103 expression), the presence of CD103+ TRMs suggests that the cell fate of this population could have been determined by the original tumor microenvironment [24]. This CD103+CD69+ TRM population was found to be more cytotoxic, expressed activation markers (PD-1, CD39, and CD49a), and was the only population that was associated with better liver metastasis outcomes [24]. This suggests that the metastasis seeding destination may matter less for TRM phenotype and functionality than the TME of origin, although much more research will be needed to support this theory. PD-1 expression and function also contribute to the heterogeneity of TRM phenotypes, with multiple reports implicating it as a marker of residency and activation rather than a marker of exhaustion [24,33,36,37]. Furthermore, characterization of skin TRMs revealed PD-1 to be crucial for early TRM engraftment and fate-determination by sensitizing TRMs to TGF-β [36]. PD-1 knockout in skin TRMs resulted in the loss of skin-specific transcriptional signatures such as extracellular matrix-related transcripts and adoption of a similar phenotype to PD-1 knockout TRMs from the spleen [36]. Likewise, anti-PD-1 blockade induced toxicity and impaired skin TRM cell formation [36], suggesting that PD-1 plays a very important role in TRM function in the skin. This highlights yet another aspect that can influence the heterogeneity of TRM phenotypes and responses within different organs and should be considered when developing skin-targeted TRM therapeutics.

While TRM cells have traditionally been defined as memory T cells that are resident within non-lymphoid tissues, recent evidence suggests that they can also establish long-term, non-recirculating residency in lymph nodes (LNs) following local antigen exposure [38,39]. Large numbers of CD8+CD103+CD69+ TRM cells have been observed in human melanoma-infiltrated lymph nodes through both immunofluorescence and transcriptional analysis, and these cells were found to concentrate at the tumor border [34]. In functional parabiosis experiments, only mice retaining these LN TRM cells—but not their partner mice sharing all circulating T cells—rejected direct challenge with melanoma metastases, demonstrating that LN TRM cells may suffice for metastatic protection in some contexts [34]. TRM cells frequently position themselves at the margins of emerging metastases. For instance, CD8+CD103+ TRMs have been found to cluster at the border of human melanoma metastases in the skin and lymph node [40]. Further, a recent case report of an 82-year-old melanoma patient demonstrated that an in-transit micrometastasis was densely infiltrated by CD8+CD103+ TRM cells that surrounded individual melanoma cells [41]. These spatial proximities imply that TRMs could enforce cancer-immune equilibrium even at the earliest, clinically undetectable stages of metastatic growth, which could be exploited for therapeutic fortification against future metastases. Still, the presence alone of TRMs does not always guarantee metastatic control: in a vaginal metastatic melanoma patient, CD8+CD103+ TRM cells clustered at the margins of distant metastases and expanded following anti-PD1 therapy but were insufficient to clear the tumors due to tumor MHC-I loss [42]. Despite this failure, in vitro experiments demonstrated the ability of these cells to recognize the metastatic tumor cells and respond with stronger cytotoxicity than all other CD8+ T cell types, suggesting that in vivo TRM cell responses are likely suppressed in some contexts [42]. The maintenance of close spatial associations between metastases and TRM cells across diverse metastatic relevant organs (such as lymph nodes, lungs, and liver) suggests a clear opportunity for anti-metastatic therapies; however, more work will be needed to verify true functional interactions (like direct cell killing) between TRM and metastases beyond just spatial proximity.

Taken together, this early evidence suggests a clear potential role for TRM cells in controlling metastasis spread, seeding, and growth. While the mechanistic understanding of how TRM cells directly operate within each tissue and respond to diverse contexts and cues is still developing, our current knowledge suggests a unique opportunity for therapeutically boosting these tissue defenders to provide early protection for highly metastatic cancers.

## 4. TRM Therapeutic and Biomarker Implications for Metastasis Control

Although direct interventional data is still emerging, multiple lines of preclinical evidence and early clinical correlations point to tissue-resident memory T (TRM) cells both as targets for anti-metastatic therapies and as powerful biomarkers of metastatic risk and response.

### 4.1. Therapeutics Avenues Targeting TRM Cell Populations

Exploiting TRM cell biology for therapeutic applications has been widely suggested in both infection and cancer contexts. While most studies to date have focused on enhancing primary-tumor control, growing evidence indicates that TRM cell-based interventions can also prevent or clear metastases. Diverse strategies to manipulate TRM cell expansion, function, and trafficking at metastases are summarized in Table 1, and explored in depth below.

Vaccine-based strategies for inducing differentiation and expansion of TRM cell populations have garnered the most interest within the field, with studies recognizing their potential as a cancer therapy [43]. Prime-boost vaccination specifically has emerged as a powerful approach to preemptively seed TRM cells to protect tissues against metastasis. One study demonstrated that tumor DNA priming immunization followed by an intranasal live-attenuated influenza boost can reprogram TCM cells into CD103+CD69+ TRM cells within two days of mucosal boosting, providing robust protection against melanoma and mesothelioma lung metastases in mice [35]. In a complementary strategy, a subcutaneous prime and intranasal boost of microsphere encapsulated tumor antigen peptides generated lung TRM cell pools detectable two months post-boost that significantly reduced B16F10 melanoma nodules in the lung [44]. Extending these principles, an intranasal adenoviral vector alongside an IL-1β adjuvant elicited high frequencies of lung TRM cells that, even in prophylactic settings, curtailed 4T1 breast cancer metastases and, when combined with focal radiotherapy, further enhanced tumor control [45]. Similarly, intranasal delivery of CpG-coated, tumor-antigen-containing nanoparticles elicited lung-resident CD8+ TRM cells that dramatically reduced 4T1 breast-cancer metastases [46]. Evidently, there are many potential technical approaches to vaccine-based strategies for boosting TRM cells; differences in potency, selectivity, and cost will need to be investigated and weighed to develop the best therapeutic platform. Whatever the mechanism, vaccine-based approaches clearly show promise for preemptively staging TRM cells to protect tissues against metastases. Despite this potential, current evidence is limited to vaccine functionality in preventing lung metastases (likely due to the intranasal delivery method) which, although highly relevant due to the lungs being a site of frequent metastasis, calls for further investigation about efficacy of these approaches in preventing metastasis in other sites such as the lymph nodes, brain, liver, and bone.
cells-14-01297-t001_Table 1Table 1TRM cell boosting therapeutic strategies for potential metastasis control.Therapy ApproachTechniquesMechanismCancer Models Key OutcomeDirect Metastasis Control EvidenceVaccinesPrime-boost vaccination [35,44]Intranasal attenuated influenza or boost following DNA vaccine priming promotes TRM migration and differentiationMetastatic murine B16F10 (melanoma) and AB1 (mesothelioma)↑ TRM cell precursors to the lung; induce TRM cell differentiation; ↑ protection against lung metastasis High (but studies currently limited to lung metastases)
Nasal mucosa vaccination [45,46]Intranasal vaccination of an adenoviral vector vaccine with IL-1β adjuvant or tumor antigen containing CpG-coated nanoparticlesMetastatic mouse breast cancer (4T1)↑ TRM cell infiltration to existing lung metastasis; prevention of metastasis; ↓ primary tumor size (*CpG-coated nanoparticles only) 
Moderate (in-depth metastatic mouse models, aligns with prime-boost vaccination findings. Studies limited to lung metastases)Chemokine & cytokinetargetingAnti-CXCL16 [15]Neutralizes intratumoral CXCL16, allowing tumor-derived TRM cell migration to lung for metastasis protectionMurine triple negative breast cancer (4T1)↑ Tumor-specific TRM cells defending non-tumor tissues;↓ metastatic tumor burden in the lungModerate (abundant conceptual support; but direct mechanistic study limited to murine metastasis models)Immune checkpoint blockade Neoadjuvant anti-PD-1 (± CTLA-4 or chemotherapy) [47,48,49]Enhances TRM cell function and supports systemic tumor-specific immunityMurine ESCC; Phase III ESCC (NCT01216527) & Phase II oral-cancer (NCT02919683) cohorts ↑ CD8⁺CD103⁺ TRM cells; delayed progression; ↓ relapses; ↑ systemic anti-tumor immunityHigh (abundant pre-clinical and clinical data)Adoptive cell therapyTGF-β-conditioned CAR-T cells [50]Programs CAR-T into the TRM cell phenotype through exposure to TGF-β ex vivoIn vitro co-culture with pancreatic cancer cells (AsPC-1)Proof-of-concept generation of CAR-TRM cells; ↑ primary tumor control; ↑ exhaustion resistanceLow (functional TRM cells; no direct metastasis data, only primary tumor control)
iPSC-derived TRM cells [51]CRISPR-edited iPSCs showing increased TRM markers and behaviorsHuman cervical cancer (SiHa)Generation of iPSC-derived TRM-like cells; ↓ primary tumor growthLow (functional TRM generation; efficacy shown against primary tumors only; no metastasis data)Symbols ‘↑’ and ‘↓’ refer to increased and decreased, respectively.

Another emerging strategy is the manipulation of chemical signals to enhance TRM cell egress from primary tumors and promote seeding into pre-metastatic niches. Modulating chemokine gradients offers a direct way to steer TRM cell trafficking to, and increase their efficacy within metastatic niches. For instance, targeting the CXCL16-CXCR6 axis allows active redirecting of TRM cell precursors into pre-metastatic niches. In a 4T1 breast cancer model, using antibodies to neutralize intratumoral CXCL16 not only liberated CXCR6+ effector-memory cells from the primary tumor but also led to their accumulation as CD103+CD69+ TRM cells in the lungs [15]. This redistribution correlated with a marked reduction in metastatic burden when compared with IgG control mice, demonstrating a potential therapeutic approach that warrants further investigation [15]. Beyond chemokine manipulation, targeting cytokines has been shown to modulate TRM cell induction. Intranasal vaccination with a Shiga-toxin B-based mucosal vector drives lung TRM cell formation in head-and-neck and lung cancer models, but concurrent blockade of TGF-β sharply reduces CD103+CD69+ TRM frequencies and eliminates vaccine efficacy, demonstrating that in vivo TGF-β signaling is essential for generating protective TRM cells [43]. This highlights TGF-β as a critical adjuvant for therapies aimed at seeding and sustaining anti-metastatic TRM cell pools. Overall, cytokines and chemokines alike will need to be considered when attempting to direct and sustain TRM cell responses at sites of metastases, and more work will be required to identify other chemical axes that could influence TRM cell therapy success.

The therapeutic potential of TRM cells in immune checkpoint inhibitors (ICI) is the most well characterized due to the surprising finding that TRM cells are the principal responders to ICI therapy in many cancers [52,53]. Neoadjuvant ICI—treating cancer patients with ICI prior to surgical intervention—shows clear promise as an approach for metastasis prevention and control and may even outperform standard adjuvant ICI therapy in metastatic cancers [47]. Mouse studies in esophageal squamous cell carcinoma (ESCC) reveal that preventative PD-1 blockade at early ESCC stages potently increases CD8+CD103+ TRM cell infiltration, delays lesion progression, and, upon re-exposure to carcinogen, maintains TRM cell colonies that mediate prolonged survival [54]. Analysis of a clinical trial of oral cancer patients treated with anti-PD-1/CTLA-4 or neoadjuvant anti-PD-1 demonstrated that early responses to neoadjuvant ICB were mediated by pre-existing TRM cells, followed by reinforcement at the tumor-site by new T cells later primed in draining LNs [49], boosting both local and systemic immunity. Enhanced combined tumor immunity of this type has consistently been suggested to be important for protection against distant metastases [49,55]. In analyses of clinical trials in esophageal squamous cancer, combined neoadjuvant chemotherapy and neoadjuvant immunotherapy show improved control of systemic tumors and consequently fewer patients developing distant metastases post-surgery when compared to patients who received just neoadjuvant chemotherapy [48,56]. In the most recent analysis, TRM cells were identified as a significant contributor to observed enhanced metastatic control in the neoadjuvant ICI-treated group and suggested as a potential prognostic factor [56]. Combined, these insights suggest that neoadjuvant ICI (with or without chemotherapy) may be one of the most promising techniques for preventing metastasis by establishing local and systemic immunity to cancer cells to enable effective therapies across a range of cancer types.

Adoptive cell therapy (ACT) approaches that seek to engineer or select for TRM-like phenotypes remain underexplored. Despite this, due to their strong cytotoxic responses, extended tissue residency, and response efficacy to ICI, TRM cells have been recognized as potential candidates for adoptive cell therapies to prevent cancer progression and spread [57]. Chimeric antigen receptor (CAR) T-cell therapy—a form of ACT in which patient T cells are genetically modified ex vivo to express a tumor-specific receptor and then reinfused—has revolutionized cancer therapy. Despite this, CAR-T modalities have shown limited success in solid tumors, suggesting that optimization for in-tissue functionality is needed. Some authors have shown that it is possible to generate TRM-like CAR-T cells by exposing patient T cells to TGF-β alone during the engineering process, marked by upregulation of CD103, tissue residency expression patterns, and adoption of a stem-like memory state [50]. Interestingly, these engineered CAR-TRM cells were found to be resistant to exhaustion, aligning with existing knowledge about TRM cell canonical functioning in vivo [50]. Another group demonstrated the generation of hypo-immunogenic, iPSC-derived CTLs that are intrinsically enriched for TRM markers like CD103 and CD69, evade host rejection, and demonstrate robust in vitro cytotoxicity and in vivo persistence in cervical cancer models [51]. This suggests that multiple potential approaches should be explored to determine the best technique for generating ACT of TRM cell-like populations, which we propose could be used for anti-metastasis therapies. Despite these promising proofs-of-concept for designing tumor-responsive TRM cell ACTs, systemic evaluation of manufacturing conditions, homing cues, and retention signals specifically in metastatic models is still very limited. Further, metastasis control studies with TRM ACT platforms have yet to be conducted, highlighting a clear gap for future study. A deeper mechanistic understanding of TRM differentiation, seeding, and functional persistence will be essential to translate TRM-based ACT into effective anti-metastasis therapies.

The significant heterogeneity of TRMs is now well established within the field, but very little is understood about how TRMs functionally differ specifically within sites of active metastases. Further, the field lacks a clear consensus on how (or even if) metastasis-associated TRMs have specific phenotypic and molecular differences when compared with those from primary tumors or even non-cancer contexts. As such, it will be very important to consider the known aspects of TRM biology, such as homing cues (e.g., CXCR6/CXCL16), retention signals (e.g., IL-15, TGF-β), and cell-to-cell interactions (e.g., PD1-PDL1) to design therapeutics. These diverse factors (and others yet to be identified) impacting TRM function must be profiled in metastatic contexts throughout major organs of interest before any TRM-focused therapy can truly be designed in an informed manner. Existing data provides sufficient rationale for the design of TRM-based therapies, but much more work is required to understand their phenotype and behavior in diverse metastatic conditions, given that, due to their significant heterogeneity, there is likely not a single TRM therapy that will work in all metastatic cancers.

### 4.2. TRM Cells as a Biomarker for Metastatic Cancer

Despite tumor-infiltrating lymphocytes being, on average, associated with improved overall survival, it is increasingly recognized that total tumor-infiltrating lymphocytes count alone does not always act as a perfect prognostic signal in many cancers, spurring interest in more specific prognostic biomarkers [58]. TRM cells have been reported to consistently correlate with better patient prognosis across colorectal, lung, oral, ovarian, and breast cancers [23,24,35,59,60]. In colorectal cancer, high TRM cell density also forecasts a lower risk of liver metastasis [25], and in gastrointestinal-derived brain metastases, TRM cell abundance associates with markedly extended post-metastasis survival and even co-recruitment of tumor-infiltrating B cells [61]. High prevalence of TRM cells in metastatic lymph nodes of esophageal squamous cell carcinoma was also found to be associated with enhanced overall survival, even when compared to high-density TRM cells in the primary tumor [62]. Higher infiltration of CD8+CD103+ into metastatic lymph nodes is also associated with a favorable prognosis in gastric cancer, as well as correlating with better outcomes after adjuvant chemotherapy [63]. This evidence suggests a functional and prognostic significance beyond raw functional TRM cell numbers and highlights the relevance of the spatial distribution of TRM cells when considering it as a prognostic factor. Beyond site-specific findings, transcriptional analyses across thousands of metastatic melanoma samples revealed that enrichment for a memory/TRM-like signature independently predicted overall survival as well as response to anti-PD-1 therapy [64].

Controversies have emerged over the validity of TRMs as true, widely applicable prognostic factors for tumor progression and metastasis control. For instance, in a study of 379 head-and-neck squamous cell carcinoma (HNSCC) cases, high density of CD8+CD103+ TRM cells in the primary tumors predicted better overall and disease-free survival, while TRM cell density in matched lymph node metastases failed to correlate with outcome, underscoring that TRM cell impact could vary by tumor site and stage [65]. Another recent study concluded that TRM cells were not the lymph node T cell population required for tumor control, and rather may simply be a good proxy for effective anti-tumor immune responses [66]. The authors suggest that this observation could stem from TGF-β signaling impairing functional response to tumors by trapping TRMs within the lymph nodes, which was found to be rescued by TGF-β signaling deletion [66]. This result aligns directly with previous mechanistic findings on the role of excessive TGF-β in impairing systemic tumor defense [19]. As such, this observation may have more to do with the specific tissue environment of tumor-draining lymph nodes (like high TGF-β concentrations), rather than speaking directly to the broad function of TRMs. Additionally, a study of non-small cell lung cancer demonstrated that TRM profiles, dynamics, and prognostic relevance change significantly with disease progression [67], reinforcing the idea that usage of TRMs as a prognostic tool could be tumor stage dependent. Together, these results suggest that the use of TRMs as prognostic markers could fail in certain environments and tumor stages, especially in contexts with strong residency signals (e.g., TGF-β) that can impair TRM function. Thus, it will be vital to more clearly establish when and where TRMs can be effectively used for cancer prognosis, and what factors can impair their proper function in controlling both metastatic and primary tumor progression.

Despite some contradictory evidence, a strong base of existing data supports TRMs as promising prognostic biomarkers for metastasis susceptibility and patient outcomes, though further human studies are needed to unravel the full mechanistic underpinnings of their protective role and limitations of their use as prognostic tools.

## 5. Future Directions

While the current literature offers a tantalizing snapshot of TRM cells’ ability to control metastatic tissue and prevent the initial establishment of metastasis, evidence is still missing at several critical points. To convert this promise into actionable therapies, we propose four short-term research priorities: (I) better characterization of TRM-metastasis behavior in organs beyond the lung; (II) identification of TRM metabolic requirements within metastatic niches; (III) direct visualization of TRM-metastasis engagement; and (IV) testing of TRM adoptive cell therapies in metastasis models.

Existing evidence for mechanistic and therapeutic roles of TRMs against metastases heavily hinges on lung metastasis models, with only a handful of studies investigating other common metastasis sites such as the brain, lymph nodes, liver, and bone. Despite this lack of functional data, TRM density correlates with better prognosis at a range of metastatic sites, such as the liver [25], brain [61], and lymph nodes [68], which highlights a clear gap for understanding why these correlations may exist at a mechanistic level. As such, we suggest that future work should better resolve organ-specific TRM programs in significant metastasis sites. For this, utilizing a more diverse set of metastasis-generating syngeneic mouse models will be necessary, such as MC38 colon carcinoma for studying liver metastases [69] or B16F10 for lymph node metastases [70]. These models could help better test whether TRMs can engraft and curb metastatic growth in other metastatic tissue contexts, while also providing platforms for therapeutic-focused perturbations of residency signals like CXCL16 or TGF-β. Additional studies should probe for what stage TRM cell fate is determined, and whether the local metastatic microenvironment or the original tumor microenvironment plays the biggest role in dictating the efficacy of TRM metastasis control.

Further, while TRM metabolic adaptations of the conditions at the primary tumor have been investigated, more research must be conducted within tissues with active metastases to assess potential metastasis-associated TRM metabolic adaptations. Highly multiplexed spatial transcriptomics can provide the necessary resolution for identifying not only metabolic adaptations of TRMs but also aid in the identification of local microenvironment architecture and cues that cause them to arise, such as cytokines, cellular contacts, and signaling events [71]. This level of profiling will also allow for the identification of functional heterogeneity within the tissue to determine which TRM populations are most relevant for response in each metastasis model. The specific understanding of how TRMs respond to local metastatic tissue cues will dictate the rational design of therapeutics targeted to defined metastasis sites and could help inform future TRM cell engineering efforts by optimizing their survival and function.

Direct in vivo visualization of TRM and metastatic cell contact also remains unmet and will be critical for translating TRM biology into therapeutic interventions. Various techniques are available and have even been optimized for use with TRMs previously in other contexts. Two-photon microscopy allows real-time intravital imaging of a diverse set of organs and has already been used to track interactions between primary melanoma cells and CD8+CD103+ TRMs in mice [40], which we suggest could be extended to visualize and confirm interactions between metastatic cells and TRMs. TRM-focused two-photon microscopy setups have also been applied for other metastasis-relevant organs (like the lungs) in viral contexts, which could be repurposed for future metastasis-specific experiments [72]. In parallel, multiplexed immunofluorescence techniques can map associations between TRM and metastases and have already been used to identify spatial associations between TRMs and metastatic cells in the skin and lymph nodes [40]. Applying these dynamic (two-photon) and static (multiplex immunofluorescence) setups across common metastatic sites will provide critical evidence for direct TRM positioning and engagement with active metastases and could further support hypotheses about TRM’s ability to control trafficking of micrometastases.

Finally, more experiments are needed to conclusively show that TRMs generated for adoptive cell therapies (ACTs) can successfully defend against metastasis progression. The limited existing evidence on TRM-based ACTs showed success with primary tumors and tumor rechallenge [50,51], but there is still no evidence supporting their use in metastatic cancer. The most pressing task will be to establish whether engineered TRMs can home, engraft, and reduce metastatic burden in well-validated metastasis models like 4T1 breast cancer or B16F10 melanoma. Demonstration of direct interaction between engineered TRM-like cells and metastatic cells using previously mentioned imaging techniques will also greatly strengthen the rationale for TRM-based ACTs as functional in vivo tools. Although the generation of TRM-based ACTs has proven both feasible and effective in primary tumors, much more evidence will be needed before these initial platforms can be supported as a realistic means to control metastases in organs of interest like the lung, brain, and lymph nodes.

## 6. Conclusions and Perspectives

TRM cells functionally combine strategic tissue residency programs with potent effector functions and immune crosstalk ability that can position them as defenders against metastatic invasion of peripheral and lymphoid tissues. Preclinical models have demonstrated reduced metastatic burden in mice using a diverse array of TRM therapeutics, including vaccines, antibodies, and TRM cell engineering, and clinical observations demonstrate TRM’s predictive power and direct involvement in metastatic cancer control. A separate argument for TRM cells in cancer/metastasis control is antigen-dependency; there are reports supporting cancer-antigen-specific responses and cancer-antigen-independent TRM responses. Considering the complex roles of TRM cells in immune regulation, careful consideration is required regarding antigen-dependent versus -independent TRM responses in cancer control and therapy.

Questions around optimal delivery, tissue-specific efficacy, and the full mechanisms of TRM cell metastasis control and prevention remain unanswered. Still, the converging landscape of mechanistic and translational knowledge offers a solid foundation for TRM cell-based metastasis therapies, which could reduce the large existing metastasis burden or even allow its prophylactic control.

## Figures and Tables

**Figure 1 cells-14-01297-f001:**
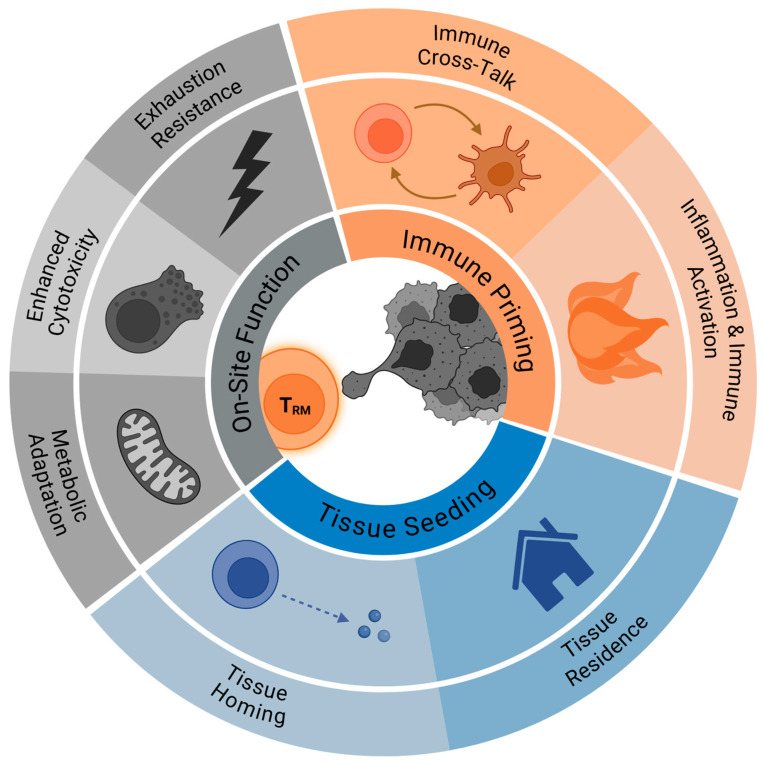
Relevant factors influencing TRM cell-mediated control of cancer metastasis.

## Data Availability

Not applicable.

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
