# Peer review of "Tissue-Resident Memory T Cells in Cancer Metastasis Control"

_cells, 2025, doi:10.3390/cells14161297_

Round 1

Reviewer 1 Report

Comments and Suggestions for Authors

This review article carefully examines how TRM cells monitor and regulate cancer metastasis. The authors begin by explaining TRM cell biology, focusing on their non-circulating, tissue-resident nature, which is marked by CD69 and CD103. The authors focus on the CXCR6-CXCL16 chemokine axis and TGF-β signaling as pathways for cell dissemination and retention in tissues, especially pre-metastatic niches. Since these molecular pathways enhance TRM accumulation in peripheral organs, they may be a suitable place to catch metastasizing tumor cells. Beyond tissue localization, TRM cells may live in nutrient-poor and hypoxic microenvironments, have tissue-adapted metabolic profiles, and respond quickly to effectors. These traits make them effective in eliminating disseminated tumor cells and initiating systemic local immune responses. The review describes how TRM cells interact with dendritic cells and other immunological components to boost anti-tumor immunity. The molecular insights section shows preclinical and clinical evidence that TRM cells prevent metastatic colonization. It emphasizes models where increasing TRM populations in distant tissues lowered metastatic burden. These treatments may work differently depending on the kind of cancer and anatomical place. The authors explore therapies to improve TRM cell activities, including vaccine-based techniques and regulation of chemokines (e.g., CXCL16), cytokines (e.g., TGF-β), and immune checkpoint inhibitors. TRM-like CAR-T or iPSC-derived T cell adoptive cell treatments (ACT) are also studied. TRM cells also improve survival and reduce metastasis in several cancers, making them useful prognostic indicators. Some exceptions exist to this pattern. The review indicates that TRM therapeutic application optimization, pre-metastatic and metastatic function differentiation, and tissue-specific TRM dynamics understanding are key knowledge gaps.

In general, the manuscript offers a comprehensive examination of the role of TRM cells in the regulation of metastasis from several aspects (biology, mechanism, therapy, and biomarker). It emphasizes numerous elements, from cellular attachment to tissue to their metabolic adaptability, primarily derived from animal studies. The article delineates several tumor forms and associated organs to substantiate its claims, such as breast cancer, melanoma, and esophageal cancer. Also, the article delineates particular therapeutic strategies: vaccinations, checkpoint inhibitors, chemokine/cytokine targeting, and adoptive cell therapy. It offers both preclinical and clinical instances, indicating that the impact of TRM cells extends beyond a laboratory phenomenon. Table 1 is exemplary, offering a systematic comparison of several TRM-centered therapy methodologies.

However, some aspects need revision.

The majority of data and therapeutic instances are for lung metastases. A comprehensive analysis of additional prevalent metastatic locations (brain, liver, bone, and lymph nodes) is absent. Please consider enhancing the article by including information on additional non-pulmonary organs or by placing greater emphasis on these deficiencies.

While it is noted that the phenotype of TRM cells may differ according to tissue and cancer type, information about the potential functional implications is insufficient. I recommend expanding the impact of TRM cell phenotypic and molecular diversity on therapeutic targeting.

Visual or in vivo evidence provides limited support for the direct associations between TRM cells and micrometastases. The authors ought to reference or promote experiments that illustrate TRM-metastasis connections through in vivo imaging techniques.

Currently, there exists only proof-of-concept data about TRM-type adoptive cell treatments, and their efficacy has not been evaluated in metastatic models. Articulate more forcefully the deficiencies in ACT development and the research trajectories required to rectify them. Information regarding the metabolic functions of TRM cells is predominantly accessible for primary tumors.

Prioritize the execution of metabolomic experiments in the metastatic context. While several data indicate the prognostic significance of TRM cells, certain research presents discrepancies that the authors address only peripherally. Explicitly contrast the conflicting results and elucidate the causes of the discrepancies (e.g., tumor type, sampling site, technical methodology).

A major revision is required. 

Reviewer 2 Report

Comments and Suggestions for Authors

The manuscript titled "Tissue-Resident Memory T cells in Cancer Metastasis Control" provides a comprehensive and timely review of the role of TRM cells in cancer metastasis, covering their biology, mechanisms of action, and therapeutic potential. The review is well-structured and synthesizes a broad range of preclinical and clinical evidence, making it a valuable resource for researchers in the field. However, there are areas where the manuscript could be improved to enhance clarity and impact.

  1. The term "TRM" is used extensively without always defining it explicitly in later sections. A brief redefinition in key sections would help maintain clarity.
  2. While the review mentions organ-specific variability in TRM function, this could be expanded. How do TRM cells differ in their anti-metastatic roles in lymph nodes versus lung or liver? A dedicated subsection or table comparing TRM phenotypes and functions across tissues would be valuable.
  3. Table 1 is useful but could include more granular details, such as clinical trial phases for human studies – if there is any.
  4. Including more recent studies (2023–2025) would ensure the review reflects the latest advancements.
  5. A short paragraph on potential pitfalls or controversies (e.g., conflicting reports on TRM cell roles in certain cancers) would provide a more balanced perspective.
  6. The conclusion outlines broad future directions but could be more specific. For instance, what are the most pressing unanswered questions about TRM cell metabolism in metastatic niches? How can spatial transcriptomics or advanced imaging techniques address these gaps?

Round 2

Reviewer 1 Report

Comments and Suggestions for Authors

Thanks for taking into account my suggestions. The revised version is now acceptable for publication.